# Within- and Between-Person Correlates of Affect and Sleep Health Among Health Science Students

**DOI:** 10.3390/brainsci14121250

**Published:** 2024-12-13

**Authors:** Yueying Wang, Jiechao Yang, Jinjin Yuan, Bilgay Izci-Balserak, Yunping Mu, Pei Chen, Bingqian Zhu

**Affiliations:** 1School of Nursing, Shanghai Jiao Tong University, Shanghai 200025, China; yue-ying.wang@connect.polyu.hk (Y.W.); yangjiechao_2017@alumni.sjtu.edu.cn (J.Y.); 2School of Nursing, Hong Kong Polytechnic University, Hong Kong SAR 999077, China; 3Songjiang Hospital Affiliated to Shanghai Jiao Tong University School of Medicine, Shanghai 201600, China; cuteolivia@sjtu.edu.cn; 4College of Nursing, University of Illinois Chicago, Chicago, IL 60612, USA; bilgay@uic.edu (B.I.-B.); pchen93@uic.edu (P.C.); 5School of Medicine, Shanghai Jiao Tong University, Shanghai 200025, China; mumu.mn@sjtu.edu.cn

**Keywords:** sleep duration, sleep quality, emotion, mixed-effect model, daily diary, health science

## Abstract

Background/Objectives: To examine the relationships between state affect and sleep health at within- and between-person levels among health science students. Methods: A correlational design was used and 54 health science students were included. The participants completed baseline and 7-day ambulatory assessments in a free-living setting. Daily sleep and affect were measured using the Consensus Sleep Diary and Positive and Negative Affect Schedule. Mixed-effect models were used to examine the effects of affect on sleep health. Results: The participants were 19.8 (*SD*, 0.6) years and 92.6% were females. Approximately 40% had poor sleep quality. Controlling for the potential confounders (e.g., age, sex, and bedtime procrastination), higher within-person negative affect predicted shorter sleep duration, lower sleep efficiency, longer sleep onset latency, and less feeling rested. Higher between-person negative affect predicted shorter sleep duration. Higher within-person positive affect predicted longer sleep onset latency. Higher within- and between-person positive affect predicted more feeling rested. Conclusions: Negative affect was most consistently associated with sleep health at the individual level. Affect regulation should be considered when delivering personalized interventions targeting sleep health among health science students.

## 1. Introduction

Sleep is an essential biobehavioral process associated with important physical and mental health outcomes [1]. Sleep health has emerged as a health-oriented concept and was defined as a multidimensional pattern of sleep–wakefulness that promotes individuals’ well-being. Good sleep health is characterized by adequate duration, high efficiency, subjective satisfaction, appropriate timing, and sustained alertness during waking hours [1]. Across the globe, sleep health among college students is vulnerable to disruption, especially health science students. Health science students may face unique challenges compared to non-health science students (e.g., heavy academic load) and thus have the highest prevalence of poor sleep compared to students of other majors [2]. Around 55% of medical and applied health students reported poor sleep quality [3]. Those disruptions in sleep health may undermine the students’ physiological and psychological well-being and impair their learning capacity and academic performance [4,5]. Based on the conceptual model of sleep health [1], various dimensions of sleep–wake function are likely to be reciprocally associated with distal health and function. Emotional well-being (e.g., affect) has been an area of interest in recent years and the sleep–emotion relationship is important for understanding the consequences of poor sleep health. Normative affect refers to normal emotions that fluctuate as a function of daily experiences [6], encompassing positive affect (PA) and negative affect (NA). These two are related, yet, distinct constructs. PA is a state of pleasurable engagement with the environment that elicits positive feelings (e.g., happiness and joy), whereas NA is characterized by feelings of distress (e.g., sadness and anger) [7]. Affect can be brief and short-term (i.e., state affect) or long-lasting and disposition-like (i.e., trait affect).

Empirical evidence suggests a bidirectional association between affect and sleep. In adolescents, those reporting higher NA had shorter sleep, and individuals had longer sleep latency following days that they had experienced high arousal PA [8]. In young women, both PA and NA were predictive of sleep (e.g., sleep quality and sleep duration) and better sleep quality was predictive of greater PA the following day [9]. In young adults, mornings with lower PA than average were associated with higher sleep efficiency that night; nights with better sleep than average were associated with higher PA the next morning [10]. Lower sleep quality ratings were associated with days characterized by less PA and more NA among older adults [11]. In healthy workers, NA did not predict subsequent sleep, but better sleep quality predicted better emotional experiences (i.e., PA) [12]. Few studies have examined the association between affect and sleep in health science students. Based on the available studies, NA was associated with sleep disturbance among college students [13,14]. Compared to non-health science students, health science students were more vulnerable to sleep disturbance [4,15]. Thus, findings from non-health science students may not be extrapolated to health science students. In addition, the Chinese education system and socio-cultural norms surrounding affect and emotional expression have unique characteristics. For instance, activated feelings associated with high-arousal PA (e.g., excitement and fun) can be valued very differently by European Americans compared with East Asians. There is a need to examine the association in the context of Asian cultures. In real life, sleep and affect (particularly state affect) both carry significant contextual fluctuations. Prior studies [13,14] were also limited by their data collection protocols where the measures were taken in one snapshot, not being able to capture the complexity of the variables of interest. In comparison, ambulatory methods (e.g., end-of-day diary) enable the collection of repeated data for the same individual in free-living settings, which could enhance external validity [16]. Current evidence has highlighted the importance of collecting and analyzing data at the participant level to account for individual biases in self-reporting over time [17]. This approach allows separating the within- and between-person effects, which may help to unveil important associations that might be masked by aggregated data.

Based on the above evidence, whether the relationship exists among health science students remains to be examined. The aim of this study was to investigate the relationships between state affect and sleep health at within- and between-person levels. We hypothesized that more NA before sleep would predict worse sleep health, whereas more PA before sleep would predict better sleep health at both the within- and between-person levels. Findings from this study will contribute to a better understanding of the important contextual role of affect in the regulation of sleep health among health science students.

## 2. Materials and Methods

### 2.1. Design

We used a correlational design to study the relationships between affect and sleep health over a 7-day period in a free-living setting.

### 2.2. Participants

This study recruited a convenience sample of college students aged 18 years or older. The participants were from a large medical university in Shanghai, China. Health science students during the first year of study could experience significant changes in their roles and lifestyles. Students during the fourth year and beyond need to complete clinical rotations and could experience disruptions in their sleep patterns. Those changes may confound study findings. Thus, only students during their second and third years of study were included. Students were excluded if they had self-reported: (1) night shift during the past month; (2) diagnosis of sleep disorders (e.g., insomnia or sleep apnea); (3) diagnosis of affective disorders. In multilevel modeling, a minimum sample size of 50 is needed to avoid biased level-2 standard errors [18]. A total of 55 participants were recruited and 54 were eligible for the final analysis.

### 2.3. Procedures

The study was approved by the Institutional Review Board. The participants were recruited through word of mouth and electronic flyers. Data were collected between October 2020 and February 2021 during which only sporadic cases of COVID were present in Shanghai. Informed consent was obtained from all the participants. The data were collected using an online platform (Wenjuanxing, version 2.0.80). Validated questionnaires were administered during the baseline assessment. During the following 7-day period (Figure 1), electronic diaries were sent by a research assistant via a social media APP (Wechat, version 7.0.22). The participants were instructed to fill out a sleep diary each morning within one hour of awakening. The sleep diary was sent between 6 and 7 a.m. and a reminder was sent if the diary was not received by noon. They were also instructed to fill out an affect diary each night before they went to bed. That diary was sent at around 7 p.m. and a reminder was sent if the diary was not received by 11 p.m.

### 2.4. Measurements

#### 2.4.1. Baseline Characteristics

The baseline characteristics of the participants were measured using a questionnaire designed by the researchers. The questionnaire included self-reported age, sex, height and weight, major (nursing and non-nursing, e.g., Clinical Medicine, Nutrition, and Pediatrics), and year of study.

#### 2.4.2. Overall Sleep Quality

The Chinese version of the Pittsburgh Sleep Quality Index (PSQI) was used to measure one’s overall sleep quality over the past month [19]. The PSQI is a self-report instrument consisting of 19 items and seven factors: subjective sleep quality, sleep duration, sleep latency, sleep efficiency, sleep disturbance, use of sleep medication, and daytime dysfunction. The PSQI global score ranges from 0 to 21, with a higher score indicating poorer sleep quality. The PSQI has good internal consistency (Cronbach’s α = 0.83). It also showed high sensitivity (89.6%), specificity (86.5%), and accuracy (88.5%) when a cut-off point was set at 5 [19]. Although PSQI was developed as a unidimensional instrument, evidence suggests that a two-factor model may perform better [20]. The Chinese version of PSQI has been validated previously [21]. The internal consistency of PSQI was moderate in this sample with a Cronbach’s α of 0.68.

#### 2.4.3. Bedtime Procrastination

The Bedtime Procrastination Scale (BPS) [22] was used to measure unhealthy sleep behavior of bedtime procrastination. It is a self-report questionnaire consisting of 9 items. Each item is scored on a five-point Likert scale: 1—(almost) never and 5—(almost) always. Items 2, 3, 7, and 9 were reverse-scored. The total BPS score is the sum of all 9 items, ranging from 9 to 45. A higher score indicates more bedtime procrastination behaviors. The BPS has been validated in the Chinese population [23]. It demonstrated good internal consistency (Cronbach’s α = 0.84) in this sample.

#### 2.4.4. Daily Sleep

The Consensus Sleep Diary for Morning (CSD-M) was used to measure daily sleep [24]. The CSD-M was developed by a panel of experts. It was derived from the core CSD which has nine questions. The CSD-M provides a more detailed assessment of sleep (e.g., early awakening and sleep quality). An example question was “What time did you get into bed?” The CSD-M was recommended as a standardized, patient-informed sleep diary [24]. Based on the conceptual model of sleep health [1], sleep health can be conceptualized from five dimensions. In this study, sleep health was operationalized as sleep duration, sleep efficiency (and onset latency), sleep timing (e.g., mid-sleep time), subjective sleep quality, and alertness (e.g., feeling rested). The former three were calculated from the diary. The latter two were measured by the following two items on the CSD-M. “How would you rate the quality of your sleep?” The response ranged from “very poor (1)” to “very good (5)”. “How rested/refreshed did you feel when you woke up for the day?” The response ranged from “not at all rested (1)” to “very well-rested (5)”. Mid-sleep time was calculated as the mid-point between sleep onset and sleep offset for each study day individually for every subject. Mid-sleep point has been used as a proxy for sleep timing [25]. In this study, the English diary was translated into Chinese by a sleep expert proficient in both English and Chinese. At the end of the diary, we added one item to ask about the “use of electronic devices after going to bed (in minutes)”.

#### 2.4.5. State Affect

Daily state PA and NA were measured using the Chinese version of the Positive and Negative Affect Schedule (PANAS) [6]. This instrument has 20 items, with 10 assessing PA (i.e., excited) and 10 assessing NA (i.e., distressed). An example item is “Right now, I felt excited”. The individuals were asked to rate the extent to which they felt each emotion using a 5-point Likert scale (1—very slightly or not at all; 5—extremely). The PA and NA scores were calculated by summing the scores of the 10 PA and 10 NA items, with a total score ranging from 10 to 50. Higher scores indicate a higher affect. The instrument has good internal consistency. The Chinese version has been validated and demonstrated good reliability in college students [26]. We calculated the Cronbach α for PA and NA during one weekday (Wednesday) and one weekend (Saturday). It ranged between 0.87 and 0.89, suggesting good internal consistency.

### 2.5. Statistical Analysis

The SPSS 22.0 (SPSS Inc., IBM, New York, NY, USA) was used for statistical analyses. Missing data, normal distribution, and outliers were checked. Among the 55 participants, one had 4-day data, one had 1 day missing on affect and one had 1 day missing on sleep. An extended sleep assessment (5 days or longer) can reduce inherent measurement errors and increase reliability [27]. Therefore, only participants who completed the diaries for 5 days or more (including at least one day of the weekend) were included in the analysis. Thus, the final analyses included 54 participants. Data were presented as mean (*SD*) or frequency (%). Person means of time-varying variables, including the use of electronic devices, affect, and sleep, were calculated for the week. Bivariate associations were checked by *t*-test or Pearson correlation analysis using person means for the week.

The repeated 7-day data were analyzed using mixed-effect models. This method could accommodate an unbalanced design (e.g., various degrees of missing data). It also enables the examination of between-person and within-person variations while accounting for the auto-correlation of multiple observations for the same individual. In order to distinguish within-person (at the individual level) and between-person (at the group level) effects, we created within- and between-person variables. Within-person variables were centered on the person mean, representing deviations from a person’s weekly average (i.e., each person’s mean subtracted from their daily value) [28]. Between-person variables were centered on the sample mean, representing deviations from the sample’s average (i.e., the sample mean subtracted from each person’s mean value). A positive value indicated a score higher than the person’s own cross-week average or a person’s score higher than others in the sample. Both the within- and between-person variables were entered into the model to test the relative contribution of each variable to the sleep outcome. The autoregressive 1 covariance matrix was used to take into account the fact that the variance of measurements taken closer in time would be more strongly associated. The final model controlled for *a priori* covariates, including age, sex, bedtime procrastination, and overall sleep quality at baseline based on previous evidence [29,30,31]. “Use of electronic devices” was added as a person-level covariate [14]. To improve statistical efficiency, baseline characteristics associated with sleep at *p* < 0.2 were also included as covariates (i.e., major) [32]. The effect size (Cohen’s d) was calculated using the estimate and *SD* which was converted from standard error. Statistical significance was set at *p* < 0.05 (two-tailed).

## 3. Results

### 3.1. Characteristics of the Sample

The baseline characteristics of the participants are shown in Table 1. The participants enrolled in this study were mainly females (92.6%), with a mean age of 19.8 years (*SD* 0.6) and a BMI of 20.5 kg/m^2^ (*SD* 2.8). The mean global PSQI was 5.7 (*SD* 2.3) and 40.7% had poor sleep quality (PSQI > 5). The mean BPS was 28.3 (*SD* 6.8). The average values of sleep and affect across the week are also presented in Table 1. Overall, the mean sleep duration per night of the whole sample was 409.7 min (*SD* 37.8), and sleep efficiency was 86.0% (*SD* 7.4). The mean PA and NA of the whole sample were 22.0 (*SD* 6.6) and 16.2 (*SD* 4.8), respectively.

### 3.2. Associations Between Sleep Health and Other Variables

Bivariate analyses were performed using the aggregated 7-day data. Associations between sleep and the baseline characteristics are shown in Table 2 and Table 3. The nursing students had a longer sleep duration (425 min vs. 399 min) and earlier mid-sleep time (03:34 vs. 04:08) than the non-nursing students. The global PSQI score was associated with all dimensions of sleep health except mid-sleep time and feeling rested. Higher BPS was associated with shorter sleep duration (*r* = −0.36, *p* < 0.01), later mid-sleep time (*r* = 0.30, *p* < 0.05), lower subjective sleep quality (*r* = −0.31, *p* < 0.05), and less feeling rested (*r* = −0.29, *p* < 0.05). Longer use of electronic devices after going to bed was associated with lower sleep efficiency (*r* = −0.82, *p* < 0.001) and longer sleep onset latency (*r* = 0.29, *p* < 0.05). PA was not associated with any of the sleep health parameters. Higher NA was associated with shorter sleep duration (*r* = −0.39, *p* < 0.01), longer sleep onset latency (*r* = 0.28, *p* < 0.05), lower subjective sleep quality (*r* = −0.28, *p* < 0.05), and less feeling rested (*r* = −0.29, *p* < 0.05).

### 3.3. Mixed-Effect Models Predicting Sleep Health from Affect

Separate mixed-effect models were run for each dimension of sleep health (Table 4). After controlling for the covariates such as demographics, overall sleep quality, bedtime procrastination, and time-varying variable (the use of electronic devices after going to bed), we found that higher NA at the within-person level predicted shorter sleep duration (estimate = −2.3, *t* = 2.88, *p* < 0.01, Cohen’s d = 0.39), lower sleep efficiency (estimate = −0.3, *t* = 3.00, *p* < 0.01, Cohen’s d = 0.41), longer sleep onset latency (estimate = 0.4, *t* = 2.00, *p* < 0.05, Cohen’s d = 0.27), and less feeling rested (estimate = −0.03, *t* = 3.00, *p* < 0.01, Cohen’s d = 0.41). Meanwhile, higher NA at the between-person level predicted shorter sleep duration (estimate = −2.5, *t* = 2.43, *p* < 0.05, Cohen’s d = 0.33).

Additionally, higher PA at the within-person level predicted longer sleep onset latency (estimate = 0.4, *p* < 0.05). Higher PA at the within- and between-person levels predicted more feeling rested (both estimates = 0.02, *p* < 0.05).

## 4. Discussion

This study was among the first that examined the within- and between-person associations between affect and sleep health among a sample of health science students in China. This study featured a 7-day ambulatory assessment and took a holistic view of sleep health. Sleep health was operationalized based on Buysse’s model [1]. We found that negative affect (NA) was a significant predictor of sleep health at the individual level. In comparison, the relationship between positive affect (PA) and sleep was not consistent. These results suggest that NA may be more consistently related to sleep than PA, particularly for the same person.

In this study, we focused on sleep health in a non-clinical young population. Approximately 41% of the participants reported poor sleep quality, comparable to the pooled estimate reported by a recent meta-analysis (40%) in medicine students [33]. The mean sleep duration was ~6.82 h, consistent with the ones found in other Asian university students [34,35]. Although the mean sleep duration of this sample was slightly lower than the recommended 7 h for this age group, their sleep efficiency was 86.0%, higher than the recommended 85%; and the sleep latency (12 min) was good (<30 min) [36]. Meanwhile, the participants’ mid-time was around 03:54, similar to a previous study [31]. Overall, sleep health in this sample is relatively good.

State NA is characterized by short-term feelings of negative emotion [7]. In this sample, higher NA than a person’s average predicted worse sleep health including shorter sleep duration, lower sleep efficiency, longer sleep onset latency, and less feeling rested upon awakening, with a small to medium effect size (Cohen’s d 0.27–0.41). At the group level, NA was only negatively associated with sleep duration. In previous studies, few have focused on health science students. Kalmbach and colleagues [9] measured 2-week daily self-reported sleep and affect in young women (mean age, 20 yrs). Different aspects of NA were uniquely predictive of sleep indices, with sadness and serenity acting as the most consistent predictors. Nonetheless, null findings have also been reported among first-year college students [37] and healthy adults [12]. Using different data collection protocols may contribute to inconsistent findings. In most of the previous studies, state affect was measured. For instance, Sin and colleagues [12] measured prior night sleep and daytime affect over an evening telephone interview. Similarly, Galambos and colleagues [37] measured sleep and affect at the end of the day. In this study, we instructed the participants to self-report their affect before they went to bed and used it to predict the subsequent sleep. It is possible that the NA experienced during the day is transient and short-lived. By the time of sleep, its impact has already disappeared. In comparison, if the participants experienced negative feelings before they went to bed, they would ruminate on them, which may impair sleep (e.g., taking a long time to fall asleep). Different data analysis protocols could also contribute to the inconsistency. In this study, we separated the within- and between-person effect, whereas aggregated data were used in the previous study [37]. It is likely that an important association (e.g., within-person association) was concealed by analyses using aggregated data. Age may be another reason for the mixed findings. Previous evidence suggests that negative emotions occur with less frequency with aging, whereas positive emotions occur with greater frequency with aging [38]. This study consisted of college students (93% female students) in their twenties. They may experience higher levels of NA and associated pre-sleep arousal than people of older age and thus exert a significant effect on sleep. Overall, our finding highlighted that NA before sleep may impair sleep health for the same person.

Positive affect is a state of pleasurable engagement with the environment that elicits positive feelings [7]. In this study, PA predicted higher feelings of being rested upon awakening at both the individual and group levels. In contrast, higher PA than a person’s average was associated with higher sleep onset latency. Both had a small effect size. de Wild-Hartmann and colleagues [29] used the daily average of affect to predict night sleep. They found that prior daytime PA was negatively associated with sleep quality, but not sleep onset latency. Moderate levels of PA may act as a buffer, dampening the potential arousal of daily hassles, whereas excessively high levels of PA may result in arousal-triggered changes in behaviors (e.g., unhealthy sleep behaviors) [39]. Prior evidence also suggests that higher PA is adaptive if it is stable over time, but not if it is highly variable [40]. The term “fragile PA” has been proposed to describe PA variability which may be a sign of maladaptive functioning [41]. Compared to stable PA, fragile PA may be harmful because it involves extreme highs and lows which have been associated with higher psychological distress [42]. It is possible that the sample had fluctuating PA that may have affected their ability to fall asleep. Based on a review of sleep and affect [43], individuals’ response and coping with daytime stressful events involve the capacity to de-arouse from active waking process which may interfere with the normal initiation of the sleep process. Individuals might need time to disengage from the arousing PA they were experiencing before bedtime, and thus it may take a long time to fall asleep. Greater PA has been associated with shorter sleep duration among high-income professionals [12]. In this study, we did not find a significant association between PA and sleep duration. This null finding could be partially due to the relatively small sample. It is also possible that sleep duration was less susceptible to PA. Additional research is needed to confirm the findings of this study.

In this study, the directionality of the association between affect and sleep health was not examined. Nevertheless, a bidirectional association likely exists. Among undergraduate students, subjective sleep satisfaction predicted PA or NA the following day [44]. Better sleep quality and longer sleep duration predicted better affect [12]. Lower sleep quality and sleep efficiency than a person’s average were associated with lower PA; lower within- and between-person sleep quality and sleep efficiency were associated with higher NA [45]. The underlying mechanism between affect and subsequent sleep remains unknown. Pathways between daily affect and sleep may differ based on the causal direction. Daily affect may impair subsequent sleep through greater emotional, physiological, or cognitive arousal (e.g., muscle tension and racing thoughts) [12], and a good night of sleep may help us to feel good and be able to cope with emotional challenges of the next day [43]. The relationship between affect and sleep is complex. Recently, a pilot study examined the impact of administering a 30-day food supplement to people with sleep mild/moderate insomnia. It was found that not only sleep quality was significantly improved, but mental well-being also showed improvement [46]. This finding suggests that addressing sleep might be beneficial for affect. Taken together, more research is needed to further illustrate the causal relationship between affect and sleep as well as the underlying pathway.

This study could advance our understanding of the relationships between affect and sleep. A major strength was that we took a holistic view of sleep health, grounded in theoretical models. Another strength was the use of a 7-day ambulatory assessment, resulting in 378 sets of measurements (54 participants × 7 days). This approach enabled us to separate the within- and between-person effects, which could provide evidence for more personalized interventions. In the statistical analyses, in addition to group-level confounders (e.g., overall sleep quality and bedtime procrastination), we also adjusted for person-level factors (i.e., the use of electronic devices). It helped to reveal the true relationship between affect and sleep health. However, there are limitations to this study. Considering response burden and participant compliance, affect was measured once per day before retiring to bed. The reciprocal relationship was not examined (i.e., whether sleep during the previous night could impact affect the following day). Future studies using a more intense measurement of affect may offer richer information. Relatedly, although the data were naturally time-lagged, this study was correlational in design. Causal inference cannot be made. Interventional studies are warranted to reveal the causal relationship. In addition, given that some dimensions of sleep (e.g., subjective satisfaction) are subjective, we only measured sleep using a sleep diary. Subjective and objective methods each offer different perspectives about sleep health and should be considered in the future. Likewise, we used feeling refreshed upon awakening as a proxy for staying alert. Future studies using other measurements may be needed (e.g., sleepiness scale). Similarly to previous studies [17,45], gender imbalance was present in this study, with most of the participants being females. In fact, in the healthcare sector, over 70% of the workforce are females, predominantly in nursing. In China, the proportion of female health professionals increased significantly from 64% in 2002 to 72% in 2020 [30]. In this study, nearly half of our participants were nursing students, which may increase the proportion of females. Although we controlled for the possible impact of sex in the analyses, additional studies with a more balanced sample are needed. Lastly, the representativeness of this sample may be limited as the sample was from one health science school in an Asian country.

## 5. Conclusions

Given that health science students are vulnerable to disruptions in their sleep health which has health consequences, understanding the predictors of their sleep is of great significance. Using an ecologically sound approach, we demonstrated that NA was most consistently associated with sleep health at the individual level. This study highlights the importance of paying attention to the correlates at the individual level and the potential impact of negative affect on sleep. Health science students will become the future workforce working at the frontline of healthcare, and their health is pivotal in providing optimal service. Our results suggest that affect regulation should be considered when delivering personalized interventions targeting sleep health among health science students, particularly at the individual level. This finding could be of interest to healthcare professionals who provide care for this population (e.g., nurses working at higher-educational institutions or sleep clinics).

## Figures and Tables

**Figure 1 brainsci-14-01250-f001:**
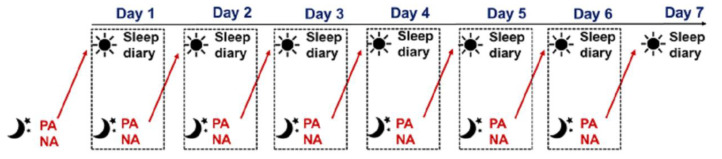
The 7-day data collection protocol. Notes. NA, negative affect; PA, positive affect.

**Table 1 brainsci-14-01250-t001:** Group-level characteristics of the participants.

Variables	Mean (*SD*)/*n* (%)	Range
Age (years)	19.8 (0.6)	18–21
Sex (female)	50 (92.6%)	
Year of study		
Second year	20 (37.0%)	
Third year	34 (63.0%)	
Major		
Nursing	22 (40.7%)	
Non-nursing	32 (59.3%)	
BMI (kg/m^2^)	20.5 (2.8)	16.6–30.7
Global PSQI	5.7 (2.3)	2–11
Poor sleep quality (PSQI > 5)	22 (40.7%)	
Bedtime Procrastination Scale	28.3 (6.8)	15–42
Average use of electronic devices (min)	29.5 (29.8)	0.0–148.3
Average sleep duration (min)	409.7 (37.8)	330.0–486.8
Average sleep efficiency (%)	86.0 (7.4)	64.7–97.1
Average sleep onset latency (min)	12.0 (9.8)	5.0–44.7
Average mid-sleep point (hh:mm)	03:54 (0:30)	02:24–4:57
Average sleep quality score	3.7 (0.5)	2.4–4.7
Average rested score	3.3 (0.6)	2.1–4.7
Average positive affect	22.0 (6.6)	10.4–36.1
Average negative affect	16.2 (4.8)	10.0–29.1

Notes: BMI, body mass index; *SD*, standard deviation; PSQI, Pittsburgh Sleep Quality Index.

**Table 2 brainsci-14-01250-t002:** Group difference in sleep health.

Predictors	Sleep Duration		Sleep Efficiency		Sleep Onset Latency		Mid-Sleep Time		Subjective Sleep Quality		Feeling Rested	
	Mean (*SD*)	*p*	Mean (*SD*)	*p*	Mean (*SD*)	*p*	Mean (*SD*)	*p*	Mean (*SD*)	*p*	Mean (*SD*)	*p*
Sex		0.33		0.12		0.66		0.16		0.10		
Female	411.1 (38.7)		85.5 (7.5)		12.2 (10.1)		03:52 (0:30)		3.7 (0.5)		3.4 (0.6)	
Male	391.8 (18.2)		91.6 (4.4)		9.9 (4.9)		04:14 (0:23)		3.2 (0.6)		2.8 (0.3)	
Year of study		0.30		0.84		0.91		0.15		0.63		0.80
2nd year	416.8 (39.1)		86.2 (7.5)		11.8 (10.5)		03:46 (0:27)		3.7 (0.6)		3.4 (0.7)	
3rd year	405.5 (37.0)		85.8 (7.5)		9.5 (1.6)		03:59 (0:30)		3.6 (0.5)		3.3 (0.6)	
Major		**0.01**		0.36		0.74		**0.03**		0.20		0.52
Non-nursing	399.0 (33.8)		85.2 (8.0)		12.4 (9.7)		04:08 (0:43)		3.6 (0.5)		3.3 (0.6)	
Nursing	425.3 (38.7)		87.1 (6.5)		11.5 (10.1)		03:43 (0:30)		3.8 (0.6)		3.4 (0.7)	

Notes: *SD*, standard deviation; bold, *p* < 0.05.

**Table 3 brainsci-14-01250-t003:** Bivariate correlations between sleep health and group-level variables.

Predictors	Sleep Duration		Sleep Efficiency		Sleep Onset Latency		Mid-Sleep Time		Subjective Sleep Quality		Feeling Rested	
	*r*	*p*	*r*	*p*	*r*	*p*	*r*	*p*	*r*	*p*	*r*	*p*
Age (years)	−0.27	0.05	−0.14	0.30	0.16	0.26	−0.01	0.95	−0.24	0.08	−0.21	0.14
BMI (kg/m^2^)	−0.13	0.36	0.04	0.76	−0.05	0.70	−0.17	0.21	0.018	0.90	−0.07	0.60
Global PSQI	−0.29	**0.04**	−0.39	**0.01**	0.31	**0.03**	0.11	0.43	−0.27	**0.046**	−0.23	0.10
BPS	−0.36	**0.01**	−0.12	0.39	0.04	0.80	0.30	**0.03**	−0.31	**0.02**	−0.29	**0.04**
Use of electronic devices (min)	−0.19	0.17	−0.82	**<0.001**	0.29	**0.03**	0.16	0.24	0.14	0.31	0.01	0.93
PA	−0.05	0.71	0.02	0.87	0.04	0.78	0.04	0.77	0.20	0.14	0.22	0.11
NA	−0.39	**0.004**	−0.21	0.12	0.28	**0.04**	0.10	0.48	−0.27	**0.048**	−0.29	**0.03**

Notes: BMI, body mass index; PSQI, Pittsburgh Sleep Quality Index; PA, positive affect; NA, negative affect; BPS, Bedtime Procrastination Scale; bold, *p* < 0.05.

**Table 4 brainsci-14-01250-t004:** Mixed-effect models predicting sleep health.

Predictors	Sleep Duration		Sleep Efficiency		Sleep Onset Latency		Mid-Sleep Time		Subjective Sleep Quality		Feeling Rested	
	Estimate (SE)	95% CI	Estimate (SE)	95% CI	Estimate (SE)	95%CI	Estimate (SE)	95% CI	Estimate (SE)	95% CI	Estimate (SE)	95% CI
Intercept	556.8 (136.6) **	285.8, 827.8	98.5 (16.1) **	66.6, 130.4	−25.4 (31.7)	−88.2, 37.3	20,547.9 (8754.0) *	3052.98, 38042.9	6.1 (1.6) **	2.8, 9.3	5.8 (1.9) **	2.03, 9.5
Age (years)	−5.1 (7.1)	−19.1, 9.02	−0.3 (0.8)	−1.9, 1.4	1.7 (1.6)	−1.6, 4.9	−538.2 (454.5)	−1446.6, 370.1	−0.1 (0.1)	−0.3, 0.1	−0.1 (0.1)	−0.3, 0.1
Sex (male)	−9.8 (17.3)	−44.2, 24.6	2.5 (2.04)	−1.5, 6.6	−1.6 (4.02)	−9.5, 6.4	1245.7 (1108.4)	−968.9, 3460.3	−0.3 (0.2)	−0.7, 0.1	−0.4 (0.2)	−0.9, 0.1
Major (non-nursing)	−18.5 (9.4) ^#^	−37.2, 0.2	0.5 (1.1)	−1.7, 2.7	−1.9 (2.2)	−6.2, 2.4	1529.2 (604.1) *	321.9, 2736.6	−0.1 (0.1)	−0.3, 0.2	0.01 (0.1)	−0.2, 0.3
Global PSQI	1.3 (2.9)	−4.4, 7.02	−0.3 (0.3)	−0.96, 0.4	1.1 (0.7)	−0.2, 2.4	−62.97 (183.02)	−428.6, 302.6	−0.04 (0.03)	−0.1, 0.03	−0.1 (0.04)	−0.1, 0.02
BPS	−1.3 (0.7) ^#^	−2.6, 0.001	−0.03 (0.08)	−0.2, 0.1	−0.1 (0.2)	−0.4, 0.2	115.1 (41.7) **	31.8, 198.5	−0.02 (0.008) ^#^	−0.03, 0.0002	−0.02 (0.01) ^#^	−0.04, 0.0001
Use of electronic devices	−0.1 (0.09)	−0.3, 0.1	−0.2 (0.01) **	−0.2, −0.1	0.1 (0.02) **	0.06, 0.1	23.1 (4.3) **	14.7, 31.5	−0.001 (0.001)	−0.003, 0.001	0.0003 (0.001)	−0.003, 0.002
Within-person PA	−0.9 (0.7)	−2.3, 0.5	−0.1 (0.1)	−0.2, 0.1	0.4 (0.2) *	0.1, 0.7	24.5 (28.7)	−31.99, 80.9	0.008 (0.008)	−0.007, 0.02	0.02 (0.01) *	0.004, 0.04
Between-person PA	0.2 (0.7)	−1.2, 1.5	0.04 (0.1)	−0.1, 0.2	0.1 (0.2)	−0.2 0.4	13.5 (42.8)	−72.1, 99.1	0.02 (0.008) ^#^	−7.5, 0.03	0.02 (0.01) *	0.004, 0.04
Within-person NA	−2.3 (0.8) **	−3.8, −0.8	−0.3 (0.1) **	−0.5, −0.1	0.4 (0.2) *	0.1, 0.7	−25.6 (30.2)	−85.1, 33.9	−0.01 (0.008)	−0.03, 0.005	−0.03 (0.01) **	−0.05, −0.01
Between-person NA	−2.5 (1.03) *	−4.6, −0.5	−0.2 (0.1)	−0.5, 0.01	0.4 (0.2)	−0.1, 0.9	−28.5 (65.8)	−160.02, 103.1	−0.02 (0.01)	−0.04, 0.006	−0.03 (0.01)	−0.05, 0.002

Notes: SE, standard error; 95% CI, 95% confidence interval; PA, positive affect; NA, negative affect; PSQI, Pittsburgh Sleep Quality Index; BPS, Bedtime Procrastination Scale; female and nursing were used as the comparator, separate models were run for each sleep variable; WP, within-person effect, representing a deviation from a person’s average; BP, between-person effect, representing person-average over the assessment period; ^#^
*p* = 0.050−0.052; * *p* < 0.05; ** *p* < 0.01.

## Data Availability

The data presented in this study are available upon request from the corresponding author due to the fact that without the permission of the research subject, any information provided by the research subject will not be disclosed to others.

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
