# Peer review of "Within- and Between-Person Correlates of Affect and Sleep Health Among Health Science Students"

_brainsci, 2024, doi:10.3390/brainsci14121250_

Round 1
Reviewer 1 Report
Comments and Suggestions for Authors
Dear Authors, congratulations on the submitted manuscript, it brings in interesting conclusions, however some changes need to be made to better emphasize the origin of this manuscript. The following are my suggestions.
1) the introduction could be split into two sub paragraphs to increase readability
2) the study design could be better represented by an illustration
3) please le view all tables, there should be a statistical error in the second one, either way all significant p values should be placed in bold.
4) in the discussion it would be useful to mention this study, read it carefully, it can give you many insights: https://doi.org/10.3390/medsci12010002
Comments on the Quality of English Languageminor editing needed
Author Response
Comments 1: The introduction could be split into two sub-paragraphs to increase readability
Response 1: We re-organized our introduction. Now it has three paragraphs, with the first one highlighting the importance of focusing on sleep health in the population of interest, the second one summarizing current evidence on the relationship between affect and sleep, and the third one introducing the aim and hypothesis.
Comments 2: The study design could be better represented by an illustration
Response 2: Figure 1 shows our research process and design. Each participant completed a 7-day data collection, during which affect was measured before bed and sleep during the previous night was assessed the following morning upon awakening.
Comments 3: Please review all tables, there should be a statistical error in the second one, either way all significant p values should be placed in bold.
Response 3: We corrected the errors and highlighted all significant P-values in bold (Pages 5-6).
Comments 4: In the discussion, it would be useful to mention this study, read it carefully, it can give you many insights: https://doi.org/10.3390/medsci12010002
Response 4: Thanks for your recommendation. We have expanded the discussion section and cited the article. (Page 9)
Reviewer 2 Report
Comments and Suggestions for Authors
Thank you for the opportunity to review this interesting and well-written manuscript which reports a daily diary study of the within- and between-person associations of affect and multidimensional sleep health. Authors report that: (1) negative affect is linked with shorter sleep duration (between-person and within-person effect) and lower sleep efficiency, longer sleep onset latency, and feeling less rested (within-person effect), and (2) positive affect is linked with feeling more rested (between-person and within-person effect) and longer sleep onset latency (within-person effect).
Although this is a generally strong paper, I believe that there are several major limitations that must be addressed:
1. I suggest that the authors should include a stronger justification for their selection of each covariate. For example, I have most often seen studies justify their covariate selection based on theory or previous observations instead of using p-values for selection.
2. Does controlling for overall sleep quality and bedtime procrastination at baseline change the way that we should interpret the sleep dependent variables? I suggest that the authors should either remove these covariates from their models or justify their inclusion and discuss how this impacts the interpretation of their results.
3. Was any consideration given to the possibility of inflated Type I error since that separate analyses were conducted for each sleep parameter? This would be important for interpreting p-values between 0.01-0.05 which would likely not remain significant if correction for multiple comparisons was applied.
I also have several smaller suggestions for improving the paper:
4. Correct font size of (Cronbach’s α) on Page 4, Line 147
5. Rephrase for conciseness and clarity on Line 39: “extremely huge academic load”
6. Add more substantive discussion of how gender may have impacted your results given that sample was 92.6% female
7. Discuss how average sleep characteristics of participants in your study compares to other studies of medical and health science students
Author Response
Comments 1: I suggest that the authors should include a stronger justification for their selection of each covariate. For example, I have most often seen studies justify their covariate selection based on theory or previous observations instead of using p-values for selection.
Response 1: Thanks for your comment. Our selection of the covariates was based on both previous evidence and the bi-variate analyses. We clarified it in the Stats section. (Page 5)
Comments 2: Does controlling for overall sleep quality and bedtime procrastination at baseline change the way that we should interpret the sleep dependent variables? I suggest that the authors should either remove these covariates from their models or justify their inclusion and discuss how this impacts the interpretation of their results.
Response 2: We added justifications for including these variables in the analyses. We also discussed it in the discussion section. (Pages 5 and 9)
Comments 3: Was any consideration given to the possibility of inflated Type I error since separate analyses were conducted for each sleep parameter? This would be important for interpreting p-values between 0.01-0.05 which would likely not remain significant if correction for multiple comparisons was applied.
Response 3: Based on previous evidence (Balancing Type I error and power in linear mixed models, http://dx.doi.org/10.1016/j.jml.2017.01.001), higher power can be achieved without an inflated Type I error rate if a model selection criterion is used in mixed-effect models. This may balance out the inflated Type I error issue.
I also have several smaller suggestions for improving the paper:
Comments 4: Correct font size of (Cronbach’s α) on Page 4, Line 147
Response 4: We corrected the font accordingly. (Page 4)
Comments 5: Rephrase for conciseness and clarity on Line 39: “extremely huge academic load”
Response 5: We changed it to “heavy academic load” (Page 1)
Comments 6: Add more substantive discussion of how gender may have impacted your results given that sample was 92.6% female
Response 6: We added more discussions on gender imbalance in the limitation section. (Page 10)
Comments 7: Discuss how average sleep characteristics of participants in your study compares to other studies of medical and health science students. For Table 2, please have the authors provide the respective times for waking up and going to bed.
Response 7: We added more relevant discussions. (Page 8) From tables 2 to 4, we included the six dimensions of sleep health. Times for waking up and going to bed can be used to calculate the sleep parameters, but are not the focus of this study. Including these two variables into the tables may confuse the readers. We thus did not include them.
Reviewer 3 Report
Comments and Suggestions for Authors
The manuscript presents the results of an interesting study on within- and between-person correlates of affect and sleep health among health science students.
· The theoretical framework is adequate, based on relevant and up-to-date literature sources. The aim of the study is clear... very specific. However, there are some aspects that if clarified would improve the understanding of the study:
· Participants. There is a limitation in the sample size, especially in the gender distribution bias... is it representative of the distribution of health science students? On the other hand, reference should be made here to the fact that the sample is not 55 participants. In fact, it is 54, although the data analysis section does specify the loss of one subject due to missing data.
· Measurements. For the Overall Sleep Quality, it would be useful to provide basic information on the factorial validity, especially because it is a questionnaire of 19 items grouped into seven factors. Information is provided on the internal consistency of the scales used using the score of the sample under study. Therefore, it would be convenient to use the McDonald omega statistic, given that it is currently recognised that it is a better fit with ordinal variables (such as Likert scale items) than Cronbach's Alpha, which is more suitable for continuous quantitative variables, and which tends to overestimate its value when applied to ordinal variables.
· Table 3. Provide effect sizes for those variables where there are significant differences (and should also provide data for the statistics. E.g. t-value, gl, ...) something that can be entered in the text (not in the table) as there are two analyses where there are significant differences. In the case of entering this information, this should also be referred to in section 2.5: Statistical Analysis
· Reference is made to the differences in the variable longer sleep duration per night than non-Nursing students, but not to the differences in Mid-sleep time.
· It is good that the significance of the correlations is pointed out. However, it should be noted that correlations lower than 0.30, even if significant, cannot be said to be strong.
· An aspect to be discussed further is the low correlation of the variables measured with the PANAS (BA and NA) with the rest of the study variables.
· Limitations related to the sample (representativeness and gender difference) are mentioned, but not to the measurement instruments, most of which are based on self-report.
Author Response
Comments 1: Participants. There is a limitation in the sample size, especially in the gender distribution bias... is it representative of the distribution of health science students? On the other hand, reference should be made here to the fact that the sample is not 55 participants. In fact, it is 54, although the data analysis section does specify the loss of one subject due to missing data
Response 1: We added more discussions on the gender imbalance in the limitation section. (Page 10) We added clarifications about the sample size in the Participants section. (Page 3)
Comments 2: Measurements. For the Overall Sleep Quality, it would be useful to provide basic information on the factorial validity, especially because it is a questionnaire of 19 items grouped into seven factors. Information is provided on the internal consistency of the scales used using the score of the sample under study. Therefore, it would be convenient to use the McDonald omega statistic, given that it is currently recognised that it is a better fit with ordinal variables (such as Likert scale items) than Cronbach's Alpha, which is more suitable for continuous quantitative variables, and which tends to overestimate its value when applied to ordinal variables.
Response 2: We added information on the factorial validity of PSQI. (Page 3) However, it is worth mentioning that we only used the global score of PSQI in the analyses. We understand emerging evidence favors the use of omega. Nonetheless, since a majority of current studies have evaluated the reliability of the scale using Cronbach's Alpha, we decided to report Cronbach's Alpha to allow for comparison with other studies.
Comments 3: Table 3. Provide effect sizes for those variables where there are significant differences (and should also provide data for the statistics. E.g. t-value, gl, ...) something that can be entered in the text (not in the table) as there are two analyses where there are significant differences. In the case of entering this information, this should also be referred to in section 2.5: Statistical Analysis
Response 3: We added the effect size of Cohen’s d and related statistics in the text. (Page 5, Page 7)
Comments 4: Reference is made to the differences in the variable longer sleep duration per night than non-Nursing students, but not to the differences in Mid-sleep time.
Response 4: We added the description of mid-sleep time in the results section. (Page 6)
Comments 5: It is good that the significance of the correlations is pointed out. However, it should be noted that correlations lower than 0.30, even if significant, cannot be said to be strong.
Response 5: We agree that the correlation was not strong and highlighted it in the discussion section. (Page 8, Page 9)
Comments 6: An aspect to be discussed further is the low correlation of the variables measured with the PANAS (BA and NA) with the rest of the study variables.
Response 6: See above.
Comments 7: Limitations related to the sample (representativeness and gender difference) are mentioned, but not to the measurement instruments, most of which are based on self-report.
Response 7: We updated our discussion on the limitations accordingly. (Page 9-10)
Round 2
Reviewer 1 Report
Comments and Suggestions for Authors
After revision, the work is ready for publication
Reviewer 2 Report
Comments and Suggestions for Authors
The authors have sufficiently addressed my original comments. Their revised paper will make an interesting contribution to the literature.
Reviewer 3 Report
Comments and Suggestions for Authors
All suggestions have been considered